# Fungi Tryptophan Synthases: What Is the Role of the Linker Connecting the α and β Structural Domains in *Hemileia vastatrix* TRPS? A Molecular Dynamics Investigation

**DOI:** 10.3390/molecules29040756

**Published:** 2024-02-06

**Authors:** Natália F. Martins, Marcos J. A. Viana, Bernard Maigret

**Affiliations:** 1EMBRAPA Agroindústria Tropical, Planalto do Pici, Fortaleza 60511-110, CE, Brazil; marcos.viana@embrapa.br; 2LORIA, UMR 7504 CNRS, Université de Lorraine, 54000 Vandoeuvre les Nancy, France; bernard.maigret@loria.fr

**Keywords:** tryptophan synthase (TRPS), fungi, *Hemileia vastatrix*, molecular dynamic

## Abstract

Tryptophan synthase (TRPS) is a complex enzyme responsible for tryptophan biosynthesis. It occurs in bacteria, plants, and fungi as an αββα heterotetramer. Although encoded by independent genes in bacteria and plants, in fungi, TRPS is generated by a single gene that concurrently expresses the α and β entities, which are linked by an elongated peculiar segment. We conducted 1 µs all-atom molecular dynamics simulations on *Hemileia vastatrix* TRPS to address two questions: (i) the role of the linker segment and (ii) the comparative mode of action. Since there is not an experimental structure, we started our simulations with homology modeling. Based on the results, it seems that TRPS makes use of an already-existing tunnel that can spontaneously move the indole moiety from the α catalytic pocket to the β one. Such behavior was completely disrupted in the simulation without the linker. In light of these results and the αβ dimer’s low stability, the full-working TRPS single genes might be the result of a particular evolution. Considering the significant losses that *Hemileia vastatrix* causes to coffee plantations, our next course of action will be to use the TRPS to look for substances that can block tryptophan production and therefore control the disease.

## 1. Introduction

Extensive research is focused on tryptophan synthase (TRPS), the terminal enzyme in the L-tryptophan synthetic pathway [1,2,3]. This pathway is crucial for the survival of various microorganisms, including bacteria, plants, and fungi [4]. Since TRPS is absent in humans, it holds great promise for the development of new classes of antibacterial drugs and herbicides. The tryptophan biosynthesis pathway is essential for bacterial growth but is absent in higher animals and humans. Therefore, drugs that can inhibit the bacterial biosynthesis of tryptophan offer a new class of antibiotics [5,6]. In plants, tryptophan synthase is an essential regulator of plant growth and plant defense [7,8]. In fungi, the tryptophan biosynthetic pathway is required for their survival, presenting a target for new classes of antifungal compounds [9].

In the majority of species, TRPS consists of a central β_2_ dimer flanked by two α-subunits, forming an αββα heterotetramer. In this regard, numerous structural studies are dedicated to understanding these enzyme mechanisms [10,11], as TRPS is a well-studied example of a complex bifunctional enzyme machinery. The overall reaction that it catalyzes is divided into two half-reactions: the first half-reaction is accomplished by an active site localized in the α subunit, while the second half-reaction is dependent on an active site in the β one. Each α active site is connected to a β active site by a ~25 Å long hydrophobic tunnel that facilitates the diffusion, through an allosteric channeling process, of the indole moiety directly to the β active site where tryptophan is finally produced.

The α-site catalysis cleaves 3-indole-D-glycerol 3′-phosphate (IGP) into indole and D-glyceraldehyde 3-phosphate (G3P). Subsequently, after the movement of the indole moiety within the α-to-β interconnecting tunnel, the β-site catalysis achieves L-tryptophan synthesis using the produced indole. The allosteric regulation of substrate channeling involves conformational changes within the subunits as well as in their quaternary states [12,13,14,15], highlighting the complex and atypical behavior of the whole system [16]. One of the main characteristics of TRPS is its structure–function conservation [17].

During the evolution of fungi [18], unlike other tryptophan synthases in which the genes for subunits are adjacent and transcribed in the order B-A, this order is reversed in fungal fusion. Here, the genes are fused in the order A-B, representing a transposition of the bacterial ones. Consequently, fungal tryptophan synthase originates from a single gene, and the protein is constructed from a continuous chain where the structural α and β subunits are fused together into a single multifunctional protein. In this protein, the part homologous to the α-chain is the N-terminal and the one related to the β chain is the C-terminal. A long linker of 20–60 amino acid residues, depending on the fungal species, connects the two α and β domains of this protein. This connector appears necessary to link the two domains, as the end of the α domain is approximately 70 Å away from the beginning of the β one.

This paper aims to help provide answers to the questions asked above using molecular dynamics simulations to investigate the structural behavior of *Hemileia vastatrix* TRPS. This work could also later help the discovery of new effective and innovative fungal TRPS inhibitors (work in progress). For that purpose, we first analyzed the capability of *Hemileia vastatrix* TRPS to have a functional mechanism similar to other bacterial TRPS types and next checked the role of the linker connecting the α and β units by removing it.

The questions are as follows: (i) is the catalytic mechanism of fungal tryptophan synthase the same as in other species, particularly concerning the tunnel between the two domains’ catalytic sites? (ii) What is the role of the supplementary linker during the allosteric phenomenon? (iii) Is a quaternary organization that mimics the one found in the αββα of other species necessary in fungal TRPS?

Unfortunately, fungal tryptophan synthases are poorly investigated, although fungi-specific tryptophan synthases (TRP1 gene) have been well characterized [19,20,21,22,23,24,25], demonstrating the vital role of this enzyme in fungal tryptophan biosynthesis and, consequently, fungal survival. Moreover, as fungi are the source of many infections, it is also interesting to investigate the behavior of typical fungal tryptophan synthases, such as the one found in *Hemileia vastatrix*, the causal agent of coffee rust [26]. *Hemileia vastatrix* is a fungus of the order Pucciniales. Its microcyclic life cycle and urediniospore shape are very different from other rust fungi [27]. The disease cycle for the parasite is a simple one: urediniospores initiate infections that develop into lesions, producing more urediniospores. Since coffee is a perennial plant with leaves that remain green throughout the year, the epidemic is continuous, with some fluctuation from season to season depending on rainfall. *Hemileia vastatrix* primarily survives as mycelium in the living tissues of the host.

## 2. Results

### 2.1. TRPS Sequence Extraction

The putative function of the previously selected protein Contig_208623_14257 was determined using BLAST [28] and InterproScan [29]. Blast alignment between this protein and the NCBI non-redundant database suggested that Contig_208623_14257 is a tryptophan synthase. Moreover, this function was supported by InterProScan, which annotated the protein as “Tryptophan synthase β subunit-like PLP-dependent enzyme” (IPR006654). However, the alignment between Contig_208623_14257 and Blast best hits revealed that this protein is truncated at the first half of the sequence (Appendix A).

To investigate whether the protein Contig_208623_14257 is truncated or misannotated, the *Puccinia graminis* [30] TRPS was aligned against the *Hemileia vastatrix* sequence using GeneWise [31], a dedicated program for protein-genome alignment. According to the GeneWise prediction, the complete *Puccinia graminis* TRPS aligned with the *Hemileia vastatrix* candidate (Appendix A). The resulting new *Hemileia vastatrix* TRP1 was successfully aligned with the five previously used TRPS types.

The TRPS sequence used in the present work is shown in Figure 1a. A phylogenetic dendrogram displays the possible relationship with other well-known fungi proteins (Figure 1b).

### 2.2. TRPS Apo 3D Structure: Homology Modeling Step

The phylogenetic tree depicted in Figure 2 clearly indicates that the 5KIN crystal structure of tryptophan synthase from *Streptococcus pneumoniae* has the closest similarity to the *H. Vastatrix* TRP1 candidate. As a result, we primarily used this template for our homology model construction. Additionally, due to finding complementary sequence correspondence between TRP1 and the 5KZM sequence from *Francisella tularensis*, we employed this structure as a secondary template. This choice was motivated by the fact that both 3D structures were ligand-free, as it has been demonstrated that the binding of small effectors or inhibitors can result in significant conformational changes [17].

The sequence alignment of α and β regions is shown in Figure 3, illustrating the conserved residues among the sequences. Initially, we aligned the 3D structures of 5KIN and 5KZM based on conserved residues. The pairwise alignment between 5KIN and TRP1 showed 45.40% identity. Next, we extracted from these PDB structures the most relevant segment, in terms of sequence identity, for the homology transfer from the templates to our TRP1 target.

The final homology model, named 3D0, is illustrated in Figure 4a,b, depicting the relative positions of the α and β regions with respect to the long linker connecting them. Structural superposition is shown in Figure 4c.

### 2.3. Target 3D Structure: MD Simulations

The conformational behavior and stability of the apo protein in the initial solvated system (Figure 5) were assessed using the usual RMSD variations and the conservation of secondary structure elements. From this analysis, it became evident that all of the α-helices and β-sheets, as obtained in the 3D0 homology models, were largely retained throughout the entire MD trajectory (Figure 6), indicating that the intra-domain topologies were maintained during the simulation.

The various RMSD plots and RMSD maps (Figure 7a,b) revealed the following: the N- and C-terminal regions of the protein were the most flexible parts, as expected. However, the overall domain/domain organization remained stable during the simulation, with the 3D model of the entire protein achieving relatively stable behavior after 600 ns. After several stabilization jumps between 1–150 ns, the α domains reached a stability plateau, presenting a weak conformational transition around 600 ns and reaching another stable state around 700 ns. The β domains appeared more unstable than the α ones, showing a stability plateau only after 300 ns, followed by a conformational transition and reaching another plateau state around 700 ns until the end of the simulation. According to the RMSD map, the two plateaus exhibited quite similar conformations. After 30 ns, the connector maintained a stable conformation at the border between the two α and β domains.

### 2.4. Domain Movements

According to the evolution of the global radius of gyration and the α-β centroids distance (Figure 8), the arrangement of the α and β domains remains quite stable after 300 ns, followed by some rearrangement of their relative position and orientation. After 700 ns, the system reaches higher stability, consistent with the RMSD observations mentioned earlier. The calculated dynamic cross-correlation map to identify correlated and anti-correlated residue displacement is presented in Figure 9. It can be observed that domains α and β are lowly (anti-) correlated to the linker. Intra-domain movements are highly correlated and anti-correlated for both domains. Inter-domain movements are mostly anti-correlated, indicating that the two domains have opposite displacements.

### 2.5. Tunnel Detection and Evolution during the MD

The tunnel between the α catalytic site (Glu50 and Asp61) and the β catalytic site (Lys378, Glu400, Thr401, Gly402, Ala403, and His406), as identified from the MD starting frame of the apo system (Figure 10), was conserved throughout the entire MD. Nevertheless, its more or less cylindrical shape exhibited fluctuations around an average radius of 2 Å, varying by 1 Å. The tunnel presented a narrow entrance and a bend around residues.

### 2.6. Channeling Process through the Tunnel

The chemical reaction begins at the α binding site by cleaving indole-3-glycerol phosphate (IGP) into indole and glyceraldehyde-3-phosphate (G3P). Subsequently, the indole is transported through the tunnel toward the β site where tryptophan is produced. We employed a molecular dynamics TMD protocol as the initial attempt to assess the capability of an indole moiety to traverse the tunnel, moving from the α domain reaction site to the β one. Various force protocols were implemented to ensure reasonably slow movement, allowing for the examination of tunnel deformations. This was achieved by utilizing an elastic constant for TMD forces of 1.8 kcal/mole/Å2. After 320 ns of simulation, the indole successfully reached the β site (Figure 11). Throughout the indole movement, we observed several flip-flop movements of the residues lining the tunnel in the β domain. These residues changed their conformations to facilitate indole crossing but returned to their original positions afterward.

Following this TMD simulation, we sought to confirm the occurrence of this transfer mechanism spontaneously after the reaction in the α site. Thus, a second simulation assessed whether the indole could spontaneously enter the tunnel and move from the α site to the β one. After 1 µs of MD, it became evident that the indole spent a significant portion of the simulation (800 ns) maneuvering around the entrance gate and only managed to cross it after this time. Figure 12a illustrates the distance variations between the indole and the α and β binding sites, revealing that the indole transfer toward the β binding site was not completed by the end of this 1 µs MD. The difficulty in indole entrance primarily stems from the narrowness of the channel gate during the initial 800 ns, maintaining a short distance between critical gating residues, especially Leu59 and Val130 (Figure 12b). The side chains of these hydrophobic residues remained closely packed together (with distances around 4 Å) during the first 800 ns, and then moved to 9 Å around 800–850 ns, allowing indole entrance, and re-packed again after indole crossing. The capability of the indole to enter the tunnel correlates with a conformational change in the β domain beginning around 600 ns, while few modifications can be observed for the α domain (Figure 12c).

The third simulation involved introducing both the indole and the G3P products of the IGP dissociation at the α site simultaneously and running another 1 µs MD using the same starting model as in the previous simulations. Analysis of the obtained trajectory indicates that the entrance of the indole into the channel corresponds to the movement of G3P in the bulk (Figure 13). Now, the indole entirely crossed the channel to reach the β site after 350 ns and then exhibited weak movement inside the β binding region. When analyzing the domain movements during indole crossing, it becomes evident that the β domain deforms as observed in the previous simulation, while the α domain remains stable.

### 2.7. No-Linker Simulation

After removing the linker, we obtained an αβ dimer for which the MD starting model was similar to the others, but now without residues 269–291. During the simulation, the relative movements of the β domain with respect to the α one were markedly different from those observed when the linker was present. When comparing the conformations obtained for the apo model, the complex model, and the unlinked model (Figure 14), it becomes apparent that in the unlinked trajectory, the domain assembly reached a vastly different organization compared to what was observed for the two others, thus raising questions about the capability of maintaining the tunnel.

In fact, when checking the possibility of keeping a stable α to β tunnel despite the removal of the linker, it becomes evident that if a tunnel still exists during the first 50 ns of the simulation, this possibility disappears quickly, mainly due to a completely different arrangement of the α and β partners. Consequently, it can be hypothesized that without the linker, the fungi tryptophan synthase machinery cannot function properly.

## 3. Discussion

The initial legitimate concern when utilizing a structural model obtained through homology modeling is to assess its quality for investigating a complex phenomenon. In the absence of experimental structural data for fungi tryptophan synthase, validation relies on comparing the model to others [14,32,33,34,35]. Alphafold-generated three-dimensional structures were available for fungi in the phylogenetic tree of Figure 2, allowing a comparison with our 3D0 model, as illustrated in Figure 15a,b. The RMSD for backbone comparisons ranged around 5 Å for the α domain and between 5 Å to 18 Å for the β domain. The lowest RMSD was observed for a fungus relatively distant from ours on the phylogenetic tree, namely *Neurospora crassa*. Despite variations in the positions of linkers, most secondary structures were conserved and similar between our model and the Alphafold-generated ones, supporting the robustness of our model for investigating *H. vastatrix* tryptophan synthase.

## 4. Materials and Methods

### 4.1. Tryptophan Synthase Gene Detection in the Hemileia vastatrix Transcript

We began with the transcriptome analysis of *H. vastatrix* previously published [36,37]. We employed a similar approach as previously outlined [38] to identify potential TRPS genes from all of the identified transcripts. Phylogenetic dendrograms were constructed for trp orthologs from bacteria and fungi in order to assess their relative evolutionary distances via bootstrapping of 1000 replications. The distances were supported by the highest bootstrap values for phylogenetic trees using Molecular Evolutionary Genetic Analysis (MEGA7) software [39].

In the absence of experimentally solved 3D structures for *H. vastatrix* tryptophan synthase, a 3D model was constructed using homology modeling. The initial step involved identifying the most similar protein templates in the PDB Protein Data Bank [40]. A total of 134 three-dimensional structures related to the keyword “tryptophan synthase” were solved via X-ray crystallography and are accessible in the PDB. These structures were associated with various species (13 different ones found in the PDB), encompassing protein wild types or mutants, in their apo forms or with bound ligands and ions, and representing the whole α-β complex or only individual α or β units.

To pinpoint the most suitable template among all of these TRPS PDB structures, we first constructed a phylogenetic dendrogram using the Phylogeny.fr web server [41], incorporating our fungi TRPS sequence and the sequences of all potential PDB templates. Once the templates were identified, the homology model was constructed using MODELLER [42] with its default settings. Loops were optimized using the MODELLER automatic loop refinement method. Before being used for the following molecular simulation steps, the quality of our 3D0 homology model was evaluated using the PROCHECK [43], H-factor [44], and ProSA-web [45] programs.

### 4.2. TRPS Target 3D Model Refinement: Molecular Dynamics Simulations (MD) Step

From the previously obtained homology model, we examined its behavior in a physiological medium. For this, we applied a molecular dynamics protocol similar to the ones we used in already published papers [46], giving us confidence about its use. The protein was embedded in a box of 150 Å^3^ with TIP3P explicit water molecules. To ensure electrostatic neutrality, 5 sodium ions were added. The NAMD program version [47] was utilized along with the CHARMM36 force field to simulate the entire system, which consisted of 325,147 atoms in the investigated system 3D0.

The MD simulations were conducted in the isobaric–isothermal ensemble, maintaining the pressure and temperature at 1 atm and 300 K, respectively, using Langevin dynamics (with a damping parameter of 1 ps^−1^) and piston approaches. The shake algorithm was implemented during the simulation. The equations of motion were integrated with a 1 fs time step, employing the r-RESPA algorithm for electrostatic forces at a slower 2 fs frequency. Long-range interactions were treated using the particle mesh Ewald approach, with an 11 Å cut-off.

The initial state for the MD was generated from the 3D0 model, refined initially using 64,000 steps of conjugate gradient minimization. Subsequently, a 10 ns MD simulation was conducted to equilibrate the entire system. A production MD run of 1 μs was then obtained, recording frames at 1 ps intervals, resulting in a total of 1,000,000 frames for subsequent analysis.

The analysis began by assessing the conservation of secondary structure elements during the MD simulation using the Timeline plugin of VMD [48]. Prior to computing the RMSD values, all frames were aligned by considering the protein backbone. RMSDs were calculated using the “RMSD trajectory tool” plugin of VMD, and an RMSD map was generated using a previously developed in-house TCL script.

### 4.3. Domain Movements

From the recorded MD trajectories of the system, a detailed analysis of protein domain movements and their potential correlation during the MD was conducted using VMD-DisRg [49]. Another method providing additional information about protein movements is dynamic cross-correlation (DCC), which illustrates the extent to which atoms move together during molecular dynamics simulations [50]. Typically, DCC tools generate NxN heatmaps, where N is the number of Cα atoms in the investigated protein, and each element corresponds to the dynamic cross-correlation between them. We utilized the BIO3D R package [51], from which a dynamic cross-correlation matrix was calculated with the dccm function. The Pearson method was employed to calculate the cross-correlation. Molecular dynamics frames were aligned on the first frame, and correlated displacements were calculated for each Cα atom.

In relation to DCC, the dynamic residue network (DRN) is useful for identifying amino-acid residues crucial for communication within the protein structure and revealing local changes in residue positions. In this approach, the protein is represented as a set of nodes and edges, where an edge (link) between two nodes (residues) is defined to have formed if the sidechain atoms approach a given distance. An existing link is assigned a scalar value of 1, while no link equals 0. Two fundamental DRN properties can be computed: average L and average BC. Residue interaction network (RIN) graphs were constructed using the NetworkX package of the MD-TASK tool [52,53]. For a protein with a total of j residues, the average shortest path index of a residue i can be determined by calculating the mean pairwise distances (D) to each residue within the protein network. Each pairwise distance (Dij) describes the path with the least number of links from all possible paths between i and j. The average shortest path (L) indicates how far down a residue is in terms of reachability from all other residues for communication, while the BC index highlights how frequently a residue participates in the shortest paths between all residue pairs. DRN reports the moving average of RIN graphs and, consequently, average L and average BC. In this case, RIN graphs were aggregated over the last 15 ns periods at time step intervals of 10 ps.

The influence of the water around the active sites and the tunnel was examined using the SSTMap program [54]. The domain/domain interface and interaction were analyzed using homemade VMD plugins Intersurf [55] and PairInt.

### 4.4. Tunnel Detection and Indole Transfer Simulation

To determine if the α-to-β interconnecting tunnel, commonly found in most tryptophan synthases, is conserved in *Hemileia vastatrix*, the tunnel’s conservation between subunits α and β of our 3D model was analyzed using the CAVER program [56]. The starting point was defined as the position of known catalytic residues in the α subunit, with the optimization of the starting point set to a maximum distance of 3 Å and a desired radius of 5 Å. Subsequently, the recognized tunnel, defined from the starting homology model, was analyzed for its evolution during the MD using CAVER Analyst [32].

Upon completion of the MD for the entire apo protein model, to investigate the indole transfer phenomenon within the tunnel, three different molecular dynamics strategies were employed.

The first strategy involved using targeted molecular dynamics (TMD) to force the transfer from the α-position to the β-binding site. In this step, the TMD facility available in NAMD was utilized, applying steering forces to guide a subset of selected atoms toward a final target moiety. At each MD time step, the RMS distance between the current selected coordinates and the target ones was computed, and several pushing forces were applied until the transfer occurred [32].

The second strategy was the simplest, involving the inclusion of the indole compound in the α-binding site of our previous model and observing its behavior during another 1 μs long MD run. The last strategy aimed to examine how the influence of IGP could enhance the indole transfer during another simulation, with the expectation that the transfer phenomenon could occur during this new MD run. The second strategy was the simplest, involving the inclusion of the indole compound in the α-binding site of our previous model and observing its behavior during another 1 μs long MD run. The last strategy aimed to examine how the influence of IGP could enhance the indole transfer during another simulation, with the expectation that the transfer phenomenon could occur during this new MD run.

### 4.5. Removing the Linker

Starting with the model built as described above, we removed the linker fragment to obtain two independent α and β subunits. Subsequently, we performed MD simulations using the same protocol as described earlier. A 1 µs MD simulation was conducted and recorded for further analysis using the same tools as previously described.

## 5. Conclusions

The workings of non-fungal tryptophan synthase channeling systems have been extensively studied in bacteria, providing a solid basis for comparison with our results. This comparison validates the observed conformational changes in the β domain after the indole moves inside the tunnel, the role of water during channeling, and the correlation between G3P expulsion in the solvent and indole crossing through the tunnel.

The role of the linker has been clearly demonstrated in our simulations for maintaining the αβ domains’ arrangement as efficiently as the more complex αββα arrangement in bacteria. The linker’s importance has already been shown in yeast tryptophan synthase for obtaining a functional enzyme. This raises evolutionary questions about tryptophan synthase: Were the α and β domains of the most distant ancestor organized as a single-chain αβ dimer or as an αββα heterotetramer? What about the stability of the αβ dimer itself in non-fungi species? Mass spectrometry experiments and computational simulations of αββα disassembly have shown that the tetramer mostly decomposes into two α subunits, one ββ dimer, and an αββ trimer, highlighting the instability of the αβ dimer. The arrangement found in a reconstructed bacterial ancestor nevertheless formed a functional heterotetrameric αββα structure. The B-A transposition order found in all bacterial genes, when applied to *Escherichia coli*, leads to a fusion protein much less active than the wild-type A–B one. Assuming that fungi appeared on Earth after bacteria, it could be hypothesized that during evolutionary processes, a gene fusion mechanism in fungi led to the creation of a single gene encoding both α and β subunits for efficient co-regulation and co-expression. Over time, mutations and recombination events occurred, developing a linker to compensate for the fragility of the B-A arrangement. As α and β subunits have complementary functions in tryptophan synthase, having both subunits within the same gene may facilitate coordinated regulation of their expression, ensuring the correct stoichiometry for proper enzyme function. This hypothesis should be validated by further experimental investigations.

## Figures and Tables

**Figure 1 molecules-29-00756-f001:**
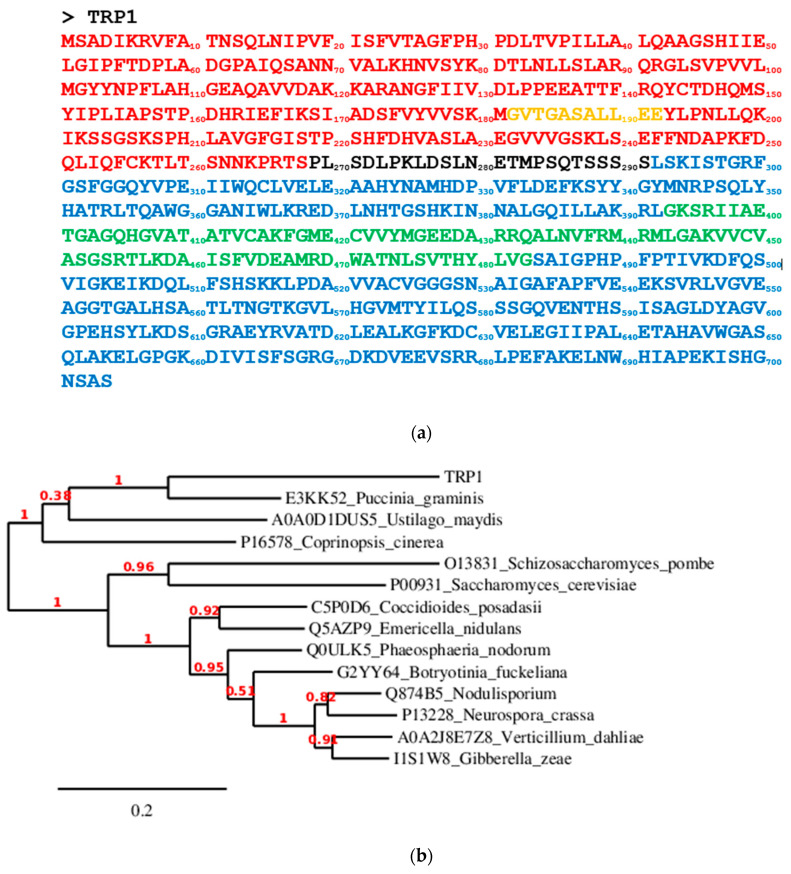
(**a**) *H. vastatrix* TRP1 tryptophan synthase sequence (Contig_208623_14257): in red, the α-subunit-related domain (residues 1–268), in blue, the β-subunit-related domain (residues 292–704), in black, the α-β connector (residues 269–291), in orange, a supplementary loop in the α domain (residues 182–192), and in green, the COMM part within the β domain (residues 393–483). (**b**) Phylogenetic dendrogram showing the proximity between the *H. vastatrix* TRP1 tryptophan synthase sequence and those from other fungi.

**Figure 2 molecules-29-00756-f002:**
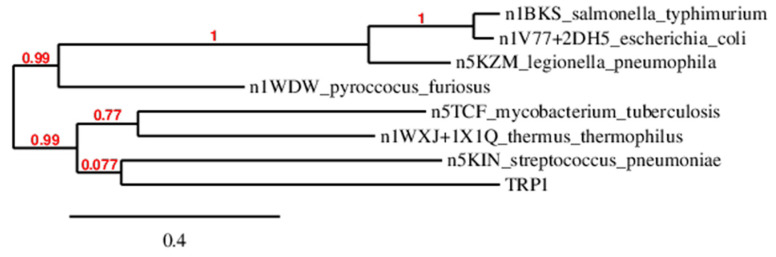
Phylogenetic dendrogram built from the TRP1 sequence and the available PDB tryptophan synthase with the maximum likelihood ratio test.

**Figure 3 molecules-29-00756-f003:**
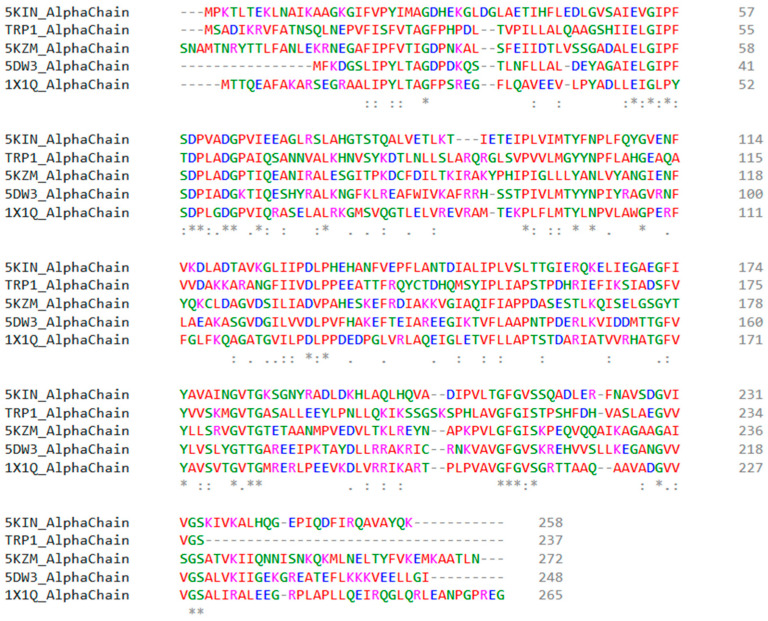
Multiple sequence alignment between the alpha and beta chains of TRP1 tryptophan synthase and the PDB templates. Physicochemical properties are represented by red (small and hydrophobic), blue (acidic), magenta (basic), and green (others).

**Figure 4 molecules-29-00756-f004:**
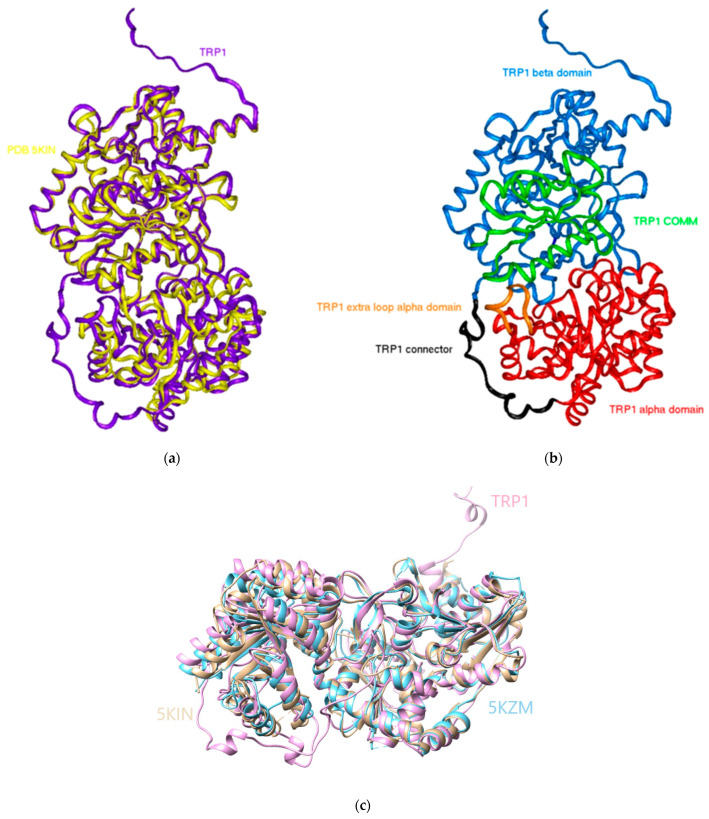
The 3D0 homology model of TRP1: (**a**) superposition with the 5KIN template, (**b**) protein structural organization, and (**c**) structural superposition between the *H. vastatrix* model and the 5KIM and 5KZM alpha and beta domains.

**Figure 5 molecules-29-00756-f005:**
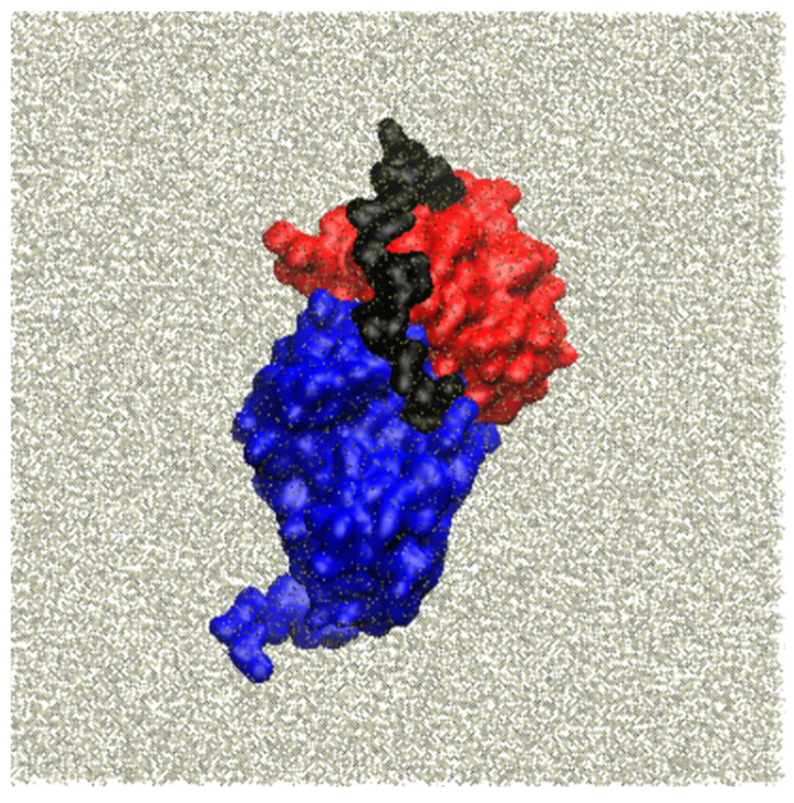
Molecular system used for the MD simulation: in red, the α domain, in blue, the β domain, and in black, the α/β connector.

**Figure 6 molecules-29-00756-f006:**
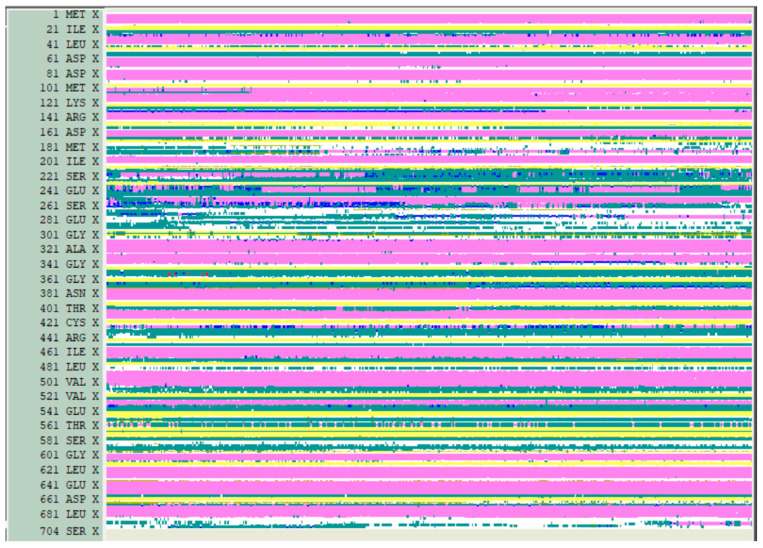
Secondary structures’ evolution during the MD simulation: helices are colored in pink and the β-sheets are colored in yellow.

**Figure 7 molecules-29-00756-f007:**
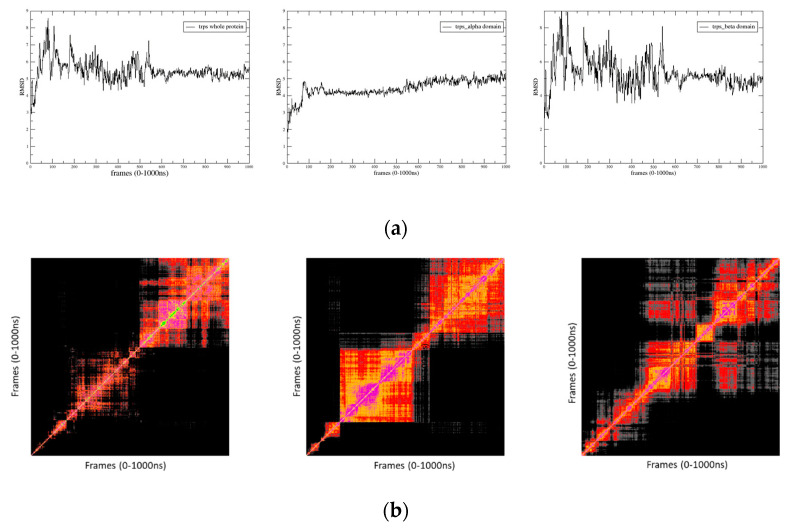
(**a**) MSD plots between the Cα carbons of the apo protein residues obtained through superposition of the MD frames to the first one along the MD trajectory. (**b**) The 2D RMSD maps between the Cα carbons of the apo protein residues obtained through superposition of each MD frame to all others along the MD trajectory.

**Figure 8 molecules-29-00756-f008:**
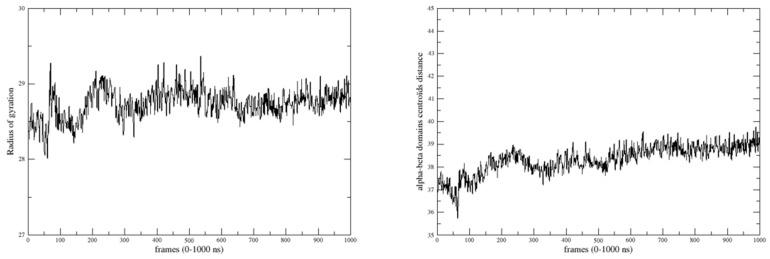
Variation in the radius of gyration of the whole protein and the domain/domain centroid distance during the 1 µs MD simulation.

**Figure 9 molecules-29-00756-f009:**
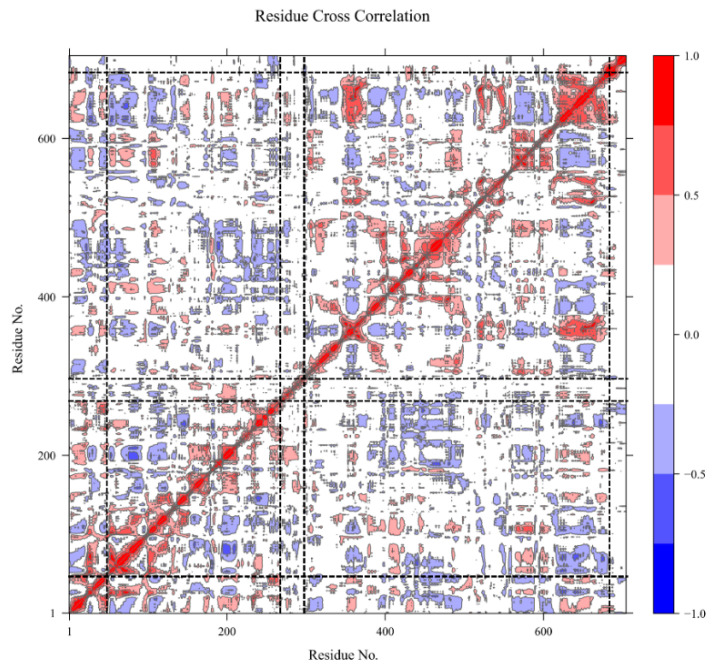
Molecular dynamics cross-correlation values of Cα atoms from the TRPS trajectory. The represented values are the Pearson cross-correlation of fluctuations between two atoms. The correlation values were calculated between −1 and 1, where 1 = complete correlation; −1 = complete anti-correlation; and 0 = no correlation. Blue values indicate anti-correlated fluctuation, and red values indicate correlated values. Dashed lines represent domain limits (50 to 270 for the alpha domain, 271 to 299 for the linker, and 300 to 650 for the beta domain).

**Figure 10 molecules-29-00756-f010:**
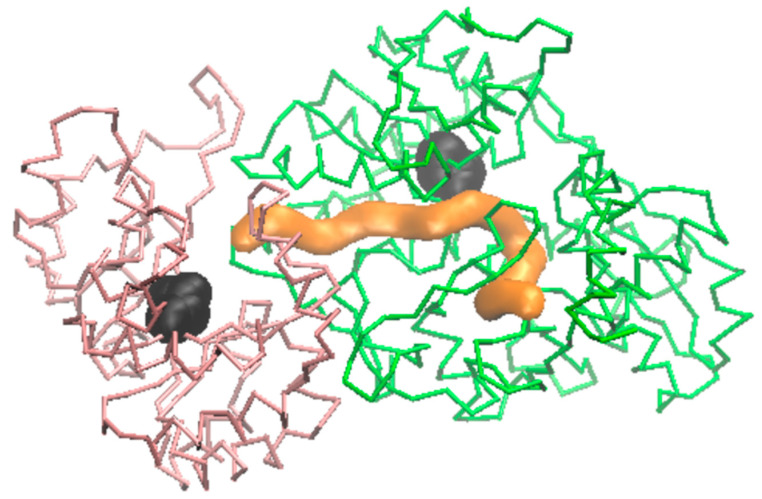
Tunnel path (orange) obtained from the α binding site to the β one (both black). The α domain trace residues 30–260 are colored pink, and the β domain 300–690 are colored green.

**Figure 11 molecules-29-00756-f011:**
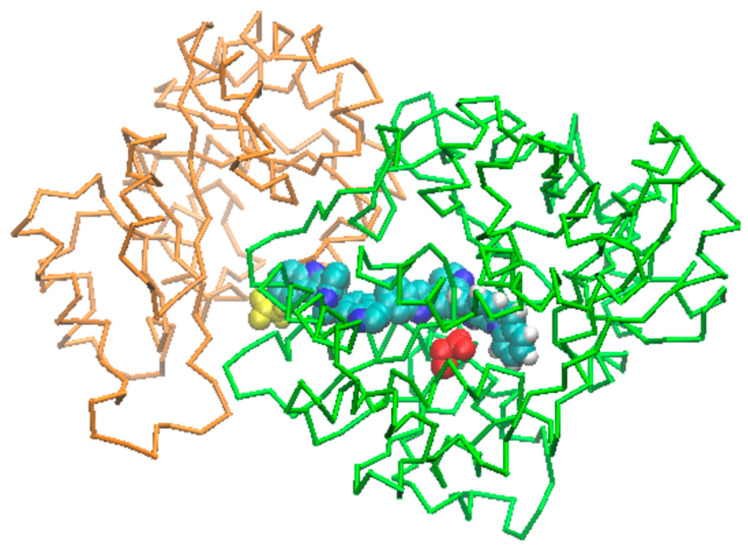
TMD jump of the indole from the α binding site to the β one by steps of 40 ns. The α domain is colored in orange and its binding site residue 61 is colored in yellow; the β one is colored in green and its binding site residue Glu400 is colored in red. The indole is colored in cyan and blue.

**Figure 12 molecules-29-00756-f012:**
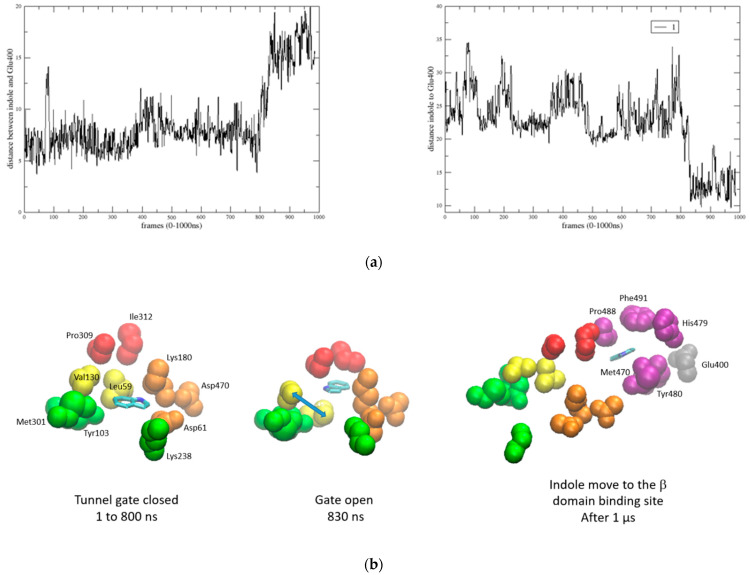
(**a**) Variation in the distances between the indole and main residues in the α and β binding sites during the 1 µS MD, respectively; (**b**) indole molecule (blue) unable to cross the tunnel start gate and blocked in the entrance by a hydrophobic cluster (yellow); (**c**) RMSD variations for the Trp1 complex with the indole.

**Figure 13 molecules-29-00756-f013:**
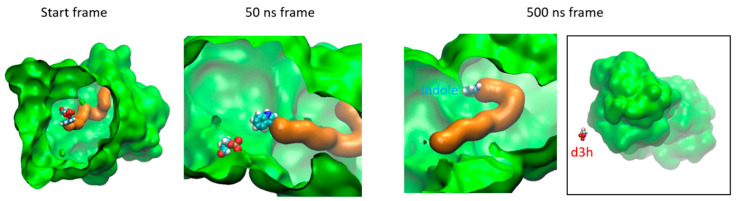
TRPS cavity surface (green) showing the position of the indole and d3h molecules (CPK colors) at the start of MD trajectory tunnel (orange).

**Figure 14 molecules-29-00756-f014:**
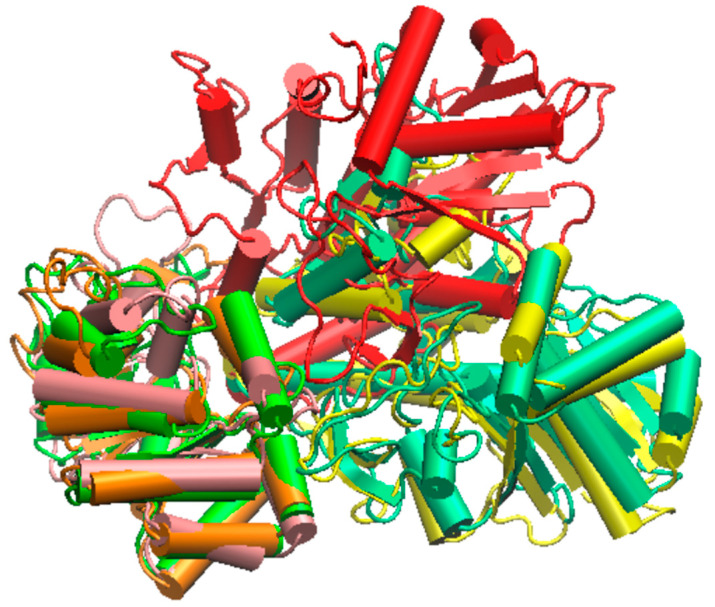
Relative position of the α and β domain as obtained after the 1 µs MDs for the apo model (α orange/β yellow), in the indole complex (green/cyan), and in the no-linker apo model (pink/red).

**Figure 15 molecules-29-00756-f015:**
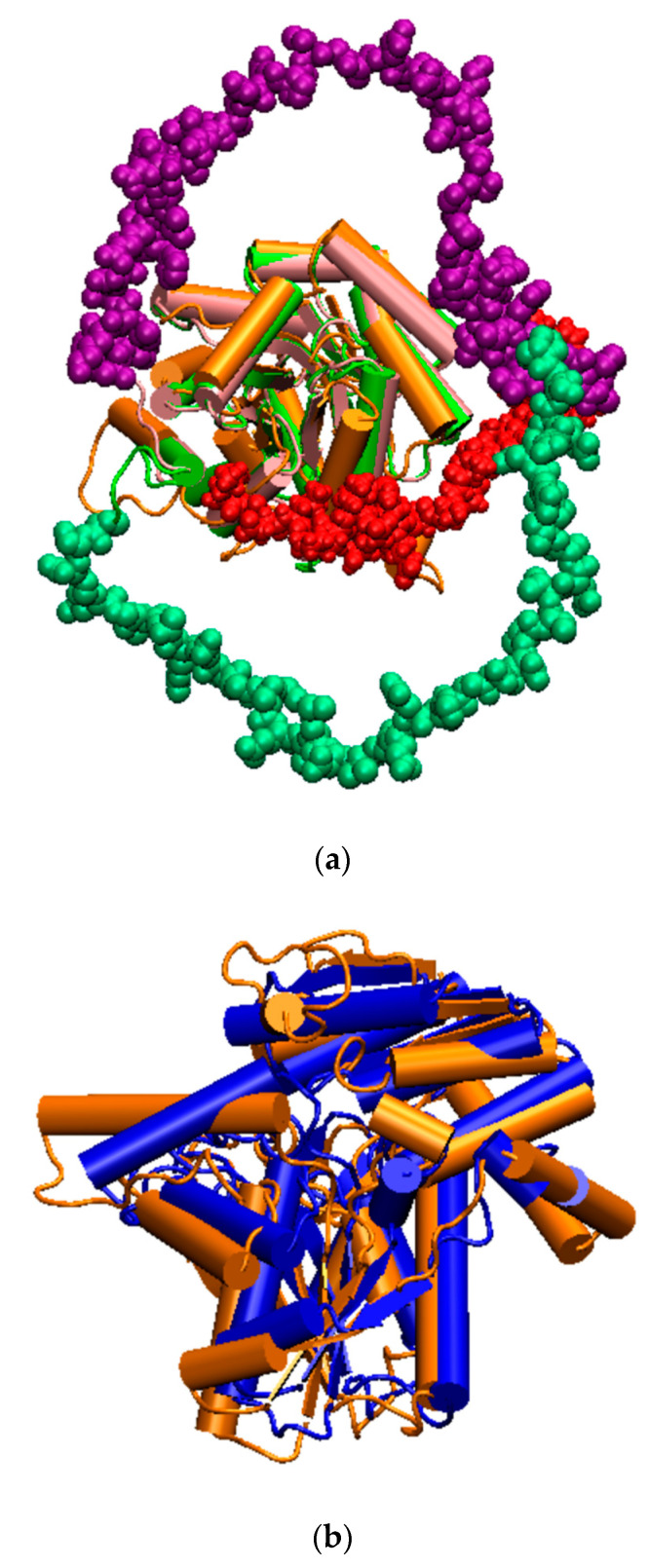
Comparison between our tryptophan synthase 3D0 model and some Alphafold ones (secondary structures drawing). (**a**) α domains and linkers: 3D0 in orange and red; *Botryotinia fuckeliana* in green and green–blue; and *Coccidioides posadas* in pink and purple; (**b**) β domains: 3D0 in orange and *Neurospora crassa* in blue.

## Data Availability

The datasets presented in this article are not readily available because the data are part of an ongoing study.

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
