# Peer review of "Fungi Tryptophan Synthases: What Is the Role of the Linker Connecting the α and β Structural Domains in Hemileia vastatrix TRPS? A Molecular Dynamics Investigation"

_molecules, 2024, doi:10.3390/molecules29040756_

Round 1

Reviewer 1 Report

Comments and Suggestions for Authors

13.01.2024

A review to evaluate its suitability for Molecules publication Type of manuscript:

Article
Title: Fungi tryptophan synthases. I - What is the role of the linker connecting
α and β structural domains in Hemileia vastatrix TRPS? A molecular dynamics investigation

Authors: Natália Florêncio Martins, Marcos J.A. Viana, Bernard Maigret

The aim of the work was to use the Tryptophan synthase (TRPS) to look for substances that can block tryptophan production and therefore control the disease.

The authors' manuscript focuses on the study of the developmental mechanism of Hemileia Vastatrix, which is the cause of coffee tree disease. For this purpose, the authors used tryptophan synthase (TRPS) enzyme and molecular modeling method to search for inhibitors of tryptophan biosynthesis to help control the disease affecting coffee plantations.

The manuscript gives the impression of a work, deeply thought out, with credibly obtained results, of a level - above average.

There are some remarks to the work, which, however, do not reduce its significance.

1.     Figure 1: does it allow confirmation and refinement of taxonomic characterisation of strains?

2.     Figure 2: How has the statistical evaluation phylogenetic tree?

3.     Figure 3: Justify the choice of the PDB model.

4.     Has the molecular modelling protocol been validated?

Respectfully, reviewer

Reviewer 2 Report

Comments and Suggestions for Authors

The work by Martins et al., advances our understating Tryptophan synthase (TRPS). The research conducted 1µs all-atom molecular dynamics simulations on Hemileia vastatrix TRPS to address two questions: (i) the role of the linker segment and (ii) the comparative mode of action. The report is well written and condensed, as well as technically appropriate for protein structure. However, before being able to recommend acceptance, I suggest authors address the following amendments.

1.      Line 68-71, Introduction part: what’s role of tryptophan synthases with bacteria and plants? Please provide this point in detail.

2.      Line 81-103, 2.1. TRPS sequence extraction: Please provide the percentage of alignment between the new Hemileia vastatrix TRP1 and with the 5 previously used TRPS.

3.      Line 105-107: Tryptophan synthase from Streptococcus pneumoniae has the closest similarity to the H. Vastatrix TRP1 candidate. How many percent identity of new Hemileia vastatrix TRP1 aligned with the 5 previously used TRPS?

4.      Line 121-124, Figure 3: What’s the meaning of color in amino acid sequence?

5.      Line 128-130, Figure 4c, “structural superposition between H. vastatrix model, 5KIM and 5KZM”. Please indicate 5KIM and 5KZM in figure 4c.

6.      Line 157, Figure 7: RMSD plots and maps between the Cα carbons of the apo protein residues. Please explain in detail this text.

7.      In references part: The manuscript does not have sufficient most recent references. The introduction should include the most recent references. A few more references of 2023 would have been better. The references of 2023 will be required as 2023 is extremely near pending the processing of the manuscript as the paper is going to be published in 2024.

Round 2

Reviewer 2 Report

Comments and Suggestions for Authors

The authors have corrected most of the suggestions.

I have agreed to publish the revised manuscript.